

# Decreased quality of life and treatment satisfaction in patients with latent autoimmune diabetes of the adult

Minerva Granado-Casas[1,2], Montserrat Martínez-Alonso[3], Nuria Alcubierre[2], Anna Ramírez-Morros[1], Marta Hernández[2,4], Esmeralda Castelblanco[1], Joan Torres-Puiggros[5,6] and Didac Mauricio[1,2]

[1] Department of Endocrinology and Nutrition, Centre for Biomedical Research on Diabetes and Associated Metabolic Diseases (CIBERDEM), Health Sciences Research Institute & University Hospital Germans Trias i Pujol, Badalona, Spain
[2] Biomedical Research Institute of Lleida, University of Lleida, Lleida, Spain
[3] Biostatistics Unit, Biomedical Research Institute of Lleida, University of Lleida, Lleida, Spain
[4] Department of Endocrinology and Nutrition, University Hospital Arnau de Vilanova, Lleida, Spain
[5] Nursing School, University of Lleida, Lleida, Spain
[6] Catalan Department of Health, Lleida, Spain

Corresponding author
Didac Mauricio,
didacmauricio@gmail.com

## ABSTRACT

**Objectives.** Our main aim was to assess the quality of life (QoL) and treatment satisfaction (TS) of subjects with LADA (latent autoimmune diabetes of the adult) and compare these measures with those of patients with other diabetes types, i.e., type 1 (T1DM) and type 2 diabetes mellitus (T2DM).

**Methods.** This was a cross-sectional study with a total of 48 patients with LADA, 297 patients with T2DM and 124 with T1DM. The Audit of Diabetes-Dependent Quality of Life (ADDQoL-19) questionnaire and the Diabetes Treatment Satisfaction Questionnaire (DTSQ) were administered. Relevant clinical variables were also assessed. The data analysis included comparisons between groups and multivariate linear models.

**Results.** The LADA patients presented lower diabetes-specific QoL ($p = 0.045$) and average weighted impact scores ($p = 0.007$) than the T2DM patients. The subgroup of LADA patients with diabetic retinopathy (DR) who were treated with insulin had a lower ADDQoL average weighted impact score than the other diabetic groups. Although the overall measure of TS was not different between the LADA and T2DM ($p = 0.389$) and T1DM ($p = 0.091$) groups, the patients with LADA showed a poorer hyperglycemic frequency perception than the T2DM patients ($p < 0.001$) and an improved frequency of hypoglycemic perception compared with the T1DM patients ($p = 0.021$).

**Conclusions.** The current findings suggest a poorer quality of life, especially in terms of DR and insulin treatment, among patients with LADA compared with those with T1DM and T2DM. Hyperglycemia frequency perception was also poorer in the LADA patients than in the T1DM and T2DM patients. Further research with prospective studies and a large number of patients is necessary.

## INTRODUCTION

Latent autoimmune diabetes in adults (LADA) is a slowly progressive form of autoimmune diabetes that presents an initial type 2 diabetes mellitus (T2DM) phenotype combined with diabetes-related autoantibodies (*Leslie et al., 2016*). At diagnosis, patients do not require insulin therapy and are often classified as T2DM patients (*Hawa et al., 2013*; *Stenstrom et al., 2005*). Patients with LADA are younger and leaner that T2DM patients (*Hawa et al., 2013*; *Stenstrom et al., 2005*). They usually have a lower body mass index (BMI), serum triglycerides (TG), waist circumference (WC), waist-to-hip ratio, and blood pressure (BP) and higher HDL cholesterol levels than the T2DM population (*Fourlanos et al., 2005*; *Hawa et al., 2009*; *Hawa et al., 2013*; *Isomaa et al., 1999*; *Mollo et al., 2013*). The cardiovascular risk profile of LADA is intermediate, falling between that of type 1 and type 2 diabetes. Additionally, these patients show lower residual endogenous insulin secretion and progress more rapidly to insulin treatment with worse glycemic control (*Fourlanos et al., 2005*; *Hawa et al., 2009*; *Isomaa et al., 1999*; *Mollo et al., 2013*; *Hernandez et al., 2015*).

Quality of life (QoL) is a subjective measure of health and well-being related to disease. It includes psychosocial features, physical functioning, mobility and personal care (*Esteban y Peña et al., 2010*). The aim of measuring QoL is to provide a more comprehensive, integral, precise and valid evaluation of patients' health status (*Speight, Reaney & Barnard, 2009*). Treatment satisfaction (TS) is an individual subjective measure that assesses the patients' experience of treatment processes and results, including ease of use, side effects and efficacy (*Hervás et al., 2002*). TS can be influenced by demographic characteristics such as age, educational level and salary (*Villar-López et al., 2009*). The development of the disease, information regarding treatment, therapy availability and costs are also associated with TS (*Villar-López et al., 2009*). Furthermore, diabetes mellitus reportedly has a negative impact QoL, but TS is optimal in this population (*Speight, Reaney & Barnard, 2009*).

Previous studies have investigated the QoL and TS of patients with type 1 diabetes mellitus (T1DM) and T2DM, but not those with LADA (*Ahola et al., 2010*; *Bradley et al., 2011*; *Depablos-Velasco et al., 2014*; *Nicolucci et al., 2009*; *Oliva, Fernandez-Bolanos & Hidalgo, 2012*). The PANORAMA study found that T2DM patients with poor glycemic control, complex treatments and severe hypoglycemic episodes showed poorer QoL compared with patients without these factors (*Bradley et al., 2011*; *Depablos-Velasco et al., 2014*). Other studies observed that QoL and treatment satisfaction were lower with increasing age, female sex, lower education level, insulin treatment and obesity, the presence of diabetic comorbidities, poorer glycemic control and lower socioeconomic status (*Ahola et al., 2010*; *Nicolucci et al., 2009*; *Oliva, Fernandez-Bolanos & Hidalgo, 2012*).

As the diagnosis of LADA among subjects with type 2 diabetes is usually delayed, which may increase the disease burden, we hypothesized that LADA patients might have a lower QoL and TS than their counterparts with T2DM; however, LADA is an autoimmune form of diabetes that usually evolves into a phenotype closely related to that of classical type 1 diabetes. Furthermore, the identification of LADA subjects may be relevant from the clinical point of view.

To our knowledge, no studies have evaluated QoL and TS in subjects with LADA. Therefore, the primary objective of this study was to assess QoL and TS in patients with LADA and to compare these measures with those of T2DM and T1DM patients. We also evaluated the factors related to both QoL and TS in the study subjects.

## MATERIALS AND METHODS

The study design was observational and cross-sectional. LADA was defined as diabetes diagnosed in individuals over 30 years of age with a positive test for glutamic acid decarboxylase (GAD) antibodies and without the need for insulin treatment during the first six months after diagnosis (*Leslie et al., 2008*; *Mollo et al., 2013*). Patients with T1DM and T2DM were diagnosed according to the current standard diagnostic criteria, as described previously (*Mollo et al., 2013*). The inclusion criteria for the three groups of patients were as follows: a diagnosis of diabetes mellitus with a disease duration of more than one year; age greater than 18 years; absence of disability due to macrovascular complications (including a history of diabetic foot disease); and absence of macroalbuminuria (defined as urine albumin-to-creatinine ratio >300 mg/g) or renal failure (estimated glomerular filtration rate <60 ml/min/1.73 m$^2$). The exclusion criteria were as follows: conditions that could affect the results, such as dementia, mental diseases, hearing and languages problems, pregnancy and disability due to cardiovascular diseases. Dementia and mental diseases had to be diagnosed by a physician and could be determined using the registered clinical records of the patient's general practitioner. No screening tools for detecting initial cognitive impairment were used. All included patients had an estimated glomerular filtration rate >60 ml/min, except for two LADA participants who had a glomerular filtration rate between 30 and 59 ml/min. Two patients in the T1DM group had previous cerebrovascular events, and one patient had bilateral femoral stenosis; none of the patients had any disability. The study was approved by the Ethics Committee of Hospital Arnau de Vilanova (Ethical Application Ref: CEIC 1079 and Ref: PI-13-095). Written informed consent was obtained from all of the subjects.

### Clinical variables

Trained researchers (MG-C and NA) conducted personal interviews with each of the patients and reviewed the medical records to collect the data regarding the variables of interest. Anthropometric measures were determined according to standard criteria. Laboratory tests and blood pressure were measured using standard procedures as previously described (*Alcubierre et al., 2014*). Hypertension and dyslipidemia were defined if the participant was undergoing pharmacological treatment for these conditions. Microalbuminuria was defined as an albumin creatinine ratio >30 mg/g. Physical activity was assessed using a validated method by *Bernstein, Morabia & Sloutskis (1999)*; participants were classified as engaging in regular exercise if they performed any physical activity that required at least ≥4 METS (The Metabolic Equivalent) of brisk walking for 30 min or more and as sedentary if they did not perform any daily physical activity or if they engaged less than 30 min of physical activity per day (*Bernstein, Morabia & Sloutskis, 1999*; *Cabrera de León et al., 2007*).

## Quality of life

QoL was assessed through the Audit of Diabetes-Dependent Quality of Life (ADDQoL-19), a disease-specific QoL questionnaire designed and validated in diabetic Spanish subjects (*Bradley et al., 2011*; *Bradley et al., 1999*; *Depablos-Velasco et al., 2014*) (File S1). This questionnaire consists of 21 items, of which 19 are related to specific life domains and are scored on a 5-point scale. The impact of diabetes on each domain is weighted according to the importance of the domain to the patient's QoL and is reported as the average weighted impact score. These scores can range from +9 (maximum positive impact) to −3 (maximum negative impact). The first two items are general and are scored separately. The first item measures current QoL and is scored from +3 (excellent) to −3 (very bad). The second item measures diabetes-specific QoL and ranges from −3 (maximum negative impact) to +1 (maximum positive impact). Moreover, five of the 19 items that may not have importance for some patients are included in a preliminary question that can be ignored if not applicable. All the questionnaires were administered individually by two trained interviewers (MG-C and NA.).

## Treatment satisfaction

TS was determined by a diabetes-specific questionnaire, the Diabetes Treatment Satisfaction Questionnaire status-version (DTSQ-s) (*Bradley, 1994*), that has been validated for the Spanish diabetic population (*Gomis et al., 2006*) (File S2). This questionnaire consists of eight items scored on a 6-point scale. The final score is weighted according to six items with total scores ranging from 36 (very satisfied) to 0 (very unsatisfied). The two remaining items measure the frequency of hyperglycemia and hypoglycemia, respectively, and are scored on a scale from 0 (never) to 6 (always).

## Sample size

To our knowledge, no previous studies have reported QoL and TS in patients with LADA. Additionally, the number of subjects with LADA is limited at the local level, and we aimed to recruit all available patients.

## Statistical analysis

The statistical analysis included the comparability among the groups of subjects with diabetes (LADA, T2DM and T1DM) and multivariate linear regression models' estimation of the variability of the overall mean score for QoL and the present QoL and diabetes-specific QoL items provided by the ADDQoL questionnaire. The TS score and the hyperglycemia and hypoglycemia frequencies were also fitted in multivariate linear regression models to identify differences between the types of diabetes after adjustment for significantly related patient characteristics. The comparison of QoL and TS between the diabetic groups was stratified by the median of the two groups. In the multivariate linear regression models, LADA patients undergoing insulin treatment were used as the reference group in all of the analyses. The comparability analysis included Pearson's chi-squared test (or Fisher's exact test in the case of any expected frequency lower than 5) to compare the distribution of qualitative characteristics. The Kruskal-Wallis test was performed to compare the distribution of quantitative characteristics, including pairwise comparisons to adjust for

multiple testing according to the Benjamini & Hochberg method; furthermore, these characteristics were described by median and interquartile intervals for each diabetic group. The significance level was set at 0.05. The statistical software R version 3.3.2 (*R Core Team, 2016*) was used for the analyses.

## RESULTS

We had previously identified a total of 106 LADA patients in the local cohort of the only reference hospital in Lleida (North-Eastern Spain). This hospital is the public reference center for specialized diabetes care for the health care district of Lleida. From this sample, 20 participants were excluded after the initial screening based on the exclusion criteria. From a sample of 86 LADA patients who were contacted to participate in the study, 51 participants initially accepted, and an additional three patients were excluded. Thus, 48 LADA patients were included in the study. From a sample of 170 T1DM patients who were contacted to participate in the study, a total of 127 agreed to participate, and 3 were excluded for pregnancy, yielding a final sample of 124 patients. As a comparison with T2DM, we used the entire group of 297 patients with T2DM who had been included in a previous study of QoL conducted at the same center (*Alcubierre et al., 2014*).

The comparison between the patients with LADA included in the study and those who refused to participate ($n = 35$) revealed no differences in clinical characteristics except for a lower frequency of insulin treatment in the non-participant group (72%; $p = 0.049$ for the comparison with the LADA patients included in the current study).

The clinical and demographic characteristics of the study groups are shown in Table 1. The LADA patients had an intermediate cardiovascular risk profile in terms of adiposity, lipids and blood pressure compared with the T2DM and T1DM groups. The patients with LADA were older than those with T1DM ($p < 0.001$). The duration of diabetes in the LADA group was longer (10.7 years) than that of the T2DM group (8 years) and shorter than that of the T1DM group (20.5 years; $p < 0.001$). The frequency of diabetic retinopathy (DR) was 22.9% in the LADA group, 49.8% in the T2DM group and 40.3% in the T1DM group ($p = 0.001$). There were no differences in glycated hemoglobin (HbA1c) among the 3 groups ($p = 0.689$).

### Quality of life

The scores for current QoL did not differ among the 3 study groups ($p = 0.503$; Table 2); however, there was a higher proportion of patients with negative diabetes-specific QoL in the LADA group (70.8%) than in the T2DM group (52.9%; $p = 0.045$). Furthermore, more subjects with LADA had a negative average weighted impact score (60.4%) compared with the T2DM patients (37.7%; $p = 0.007$). Concerning these QoL measures, we could not find differences between the LADA and T1DM patients.

The multivariate linear model of the diabetes-specific QoL score revealed a significant interaction with insulin treatment, as indicated in Table 3. The results are expressed using the group of insulin-treated patients with LADA as the reference group. The LADA and T2DM subjects without insulin treatment showed a higher diabetes-specific QoL than the reference group ($p = 0.004$ and $p < 0.001$, respectively), whereas there were no differences
**Table 1  Clinical characteristics of the study groups.**

| Characteristics | LADA (N = 48) | T2DM (N = 297) | T1DM (N = 124) | p-value[*] | p-LADA vs. T2DM | p-LADA vs. T1DM |
|---|---|---|---|---|---|---|
| Age (years) | 62 [53.8;70.2] | 60 [52.0;68.0] | 46 [37.0;53.0] | <0.001 | 0.118 | <0.001 |
| Sex, male | 26 (54.2) | 151 (50.8) | 57 (46.0) | 0.543 | | |
| Education level | | | | <0.001 | 0.181 | 0.012 |
|   <Primary | 3 (6.3) | 38 (12.8) | 17 (13.7) | | | |
|   Primary | 24 (50.0) | 169 (56.9) | 31 (25.0) | | | |
|   Secondary | 18 (37.5) | 69 (23.2) | 54 (43.5) | | | |
|   Graduate or higher | 3 (6.3) | 21 (7.1) | 22 (17.7) | | | |
| Smoking | | | | 0.891 | | |
|   Non-smoker | 24 (50.0) | 139 (47.3) | 57 (46.0) | | | |
|   Smoker, current | 11 (22.9) | 62 (21.1) | 31 (25.0) | | | |
|   Smoker, former | 13 (27.1) | 93 (31.6) | 36 (29.0) | | | |
| Physical activity | | | | 0.052 | | |
|   Sedentary | 13 (27.1) | 114 (38.4) | 34 (27.4) | | | |
|   Regular physical activity | 35 (72.9) | 183 (61.6) | 90 (72.6) | | | |
| Diabetes duration (years) | 10.7 [6.5;16.7] | 8 [4.0;15.0] | 20.5 [14.0;30.2] | <0.001 | 0.035 | <0.001 |
| BMI (kg/m$^2$) | 27.1 [24.1;30.4] | 30.6 [28.1;34.7] | 24.6 [22.5;27.2] | <0.001 | <0.001 | 0.001 |
| Waist (centimeters) | 96.7 ± 15.5 | 106 ± 11.8 | 87.7 ± 12.8 | <0.001 | <0.001 | <0.001 |
| Hypertension | 26 (54.2) | 168 (56.6) | 41 (33.1) | <0.001 | 0.878 | 0.027 |
| Systolic blood pressure (mmHg) | 134 [124.0;148.0] | 139 [127.0;150.0] | 127 [113.0;139.0] | <0.001 | 0.220 | 0.017 |
| Diastolic blood pressure (mmHg) | 73 [69.0;79.0] | 77 [70.0;84.0] | 73 [65.0;78.2] | <0.001 | 0.126 | 0.235 |
| Dyslipidemia | 34 (70.8) | 131 (44.1) | 55 (44.4) | 0.002 | 0.003 | 0.005 |
| Diabetic retinopathy | 11 (22.9) | 148 (49.8) | 50 (40.3) | 0.001 | 0.003 | 0.075 |
| Microalbuminuria | 11 (23.4) | 43 (14.5) | 11 (9.1) | 0.051 | | |
| Insulin treatment | 43 (89.6) | 97 (32.7) | 124 (100.0) | <0.001 | <0.001 | 0.001 |
| HbA1c (%) | 7.5 [6.9;8.2] | 7.6 [6.8;8.5] | 7.6 [7.0;8.1] | 0.689 | | |
| Total cholesterol (mg/dL) | 176 [155.0;202.0] | 181 [163.0;205.0] | 182 [165.0;202.0] | 0.553 | | |
| HDL-c (mg/dL) | 58.5 [40.8;70.2] | 48 [41.8;59.0] | 63 [53.9;74.0] | <0.001 | 0.040 | 0.011 |
| LDL-c (mg/dL) | 101 [84.0;123.0] | 106 [87.2;128.0] | 102 [89.6;116.0] | 0.391 | | |
| Triglycerides (mg/dL) | 91.5 [66.8;134.0] | 117 [83.0;167.0] | 65.5 [53.8;81.2] | <0.001 | 0.003 | <0.001 |

**Notes.**
[*]p-value for comparison between groups.
LADA, latent autoimmune diabetes of adult; T2DM, type 2 diabetes mellitus; T1DM, type 1 diabetes mellitus; BMI, body mass index; HbA1c, glycated haemoglobin; HDL-c, high-density lipoprotein cholesterol; LDL-c, low-density lipoprotein cholesterol.
Data are median [interquartile], $n$ (%) or means ± SD.

between the patients with T1DM and insulin-treated T2DM patients. Additionally, the presence of hypertension, longer disease duration and a larger waist circumference had a negative impact on diabetes-specific QoL.

Concerning the ADDQoL average weighted impact score, we found a significant interaction between DR, the study group (i.e., type of diabetes) and insulin treatment (Table 4). Insulin-treated LADA subjects showed a poorer average weighted impact score than their corresponding type 2 non-insulin-treated counterparts ($p < 0.001$). Additionally, T2DM patients with DR and with ($p = 0.01$) or without ($p = 0.03$) concomitant insulin

**Table 2  Descriptive analysis for the Audit of Diabetes Dependent Quality of Life (ADDQoL) results of the study groups.**

| Items | LADA (N = 48) n (%) | T2DM (N = 297) n (%) | T1DM (N = 124) n (%) | p-value* | p-LADA vs. T2DM | p-LADA vs. T1DM |
|---|---|---|---|---|---|---|
| Present QoL | | | | 0.503 | | |
| [−3,2) | 42 (87.5) | 242 (81.5) | 99 (79.8) | | | |
| [2,3] | 6 (12.5) | 55 (18.5) | 25 (20.2) | | | |
| Diabetes-specific QoL | | | | <0.001 | 0.045 | 0.634 |
| [−3,0) | 34 (70.8) | 157 (52.9) | 94 (75.8) | | | |
| [0,2] | 14 (29.2) | 140 (47.1) | 30 (24.2) | | | |
| Average weighted impact score | | | | <0.001 | 0.007 | 0.069 |
| [−6.526,−0.842) | 29 (60.4) | 112 (37.7) | 94 (75.8) | | | |
| [−0.842,0.316] | 19 (39.6) | 185 (62.3) | 30 (24.2) | | | |

Notes.

The groups are stratified by medians.

*p-value for comparison between groups.

QoL, quality of life; LADA, latent autoimmune diabetes of adult; T2DM, type 2 diabetes mellitus; T1DM, type 1 diabetes mellitus.

**Table 3  Multivariate linear regression for the Audit of Diabetes Dependent Quality of Life (ADDQoL) diabetes-specific QoL score.**

| Coefficients | Estimate | Standard error | p value |
|---|---|---|---|
| Intercept | 0.278998 | 0.407968 | 0.490 |
| T2DM* without insulin | 0.815249 | 0.154823 | <0.001 |
| LADA* without insulin | 1.187429 | 0.415424 | 0.004 |
| T2DM* insulin | 0.238438 | 0.168811 | 0.160 |
| T1DM | 0.024457 | 0.164031 | 0.880 |
| Hypertension | −0.310550 | 0.089091 | 0.001 |
| HbA1c | −0.070135 | 0.035407 | 0.050 |
| Disease duration | −0.010606 | 0.004989 | 0.030 |
| Waist circumference | −0.008147 | 0.003458 | 0.020 |

Notes.

Multiple R-squared: 0.2284 (27 cases with missing information for any variable in the model).

Reference group: LADA patients receiving insulin treatment.

*Indicates the existence of interactions between variables.

LADA, latent autoimmune diabetes of adult; T2DM, type 2 diabetes mellitus; T1DM, type 1 diabetes mellitus; HbA1c, glycated haemoglobin.

treatment and T1DM patients with DR ($p = 0.03$) had a better average weighted impact score than LADA patients undergoing insulin treatment. The presence of DR, longer disease duration, lower education level (less than a primary education) and former smoking had a negative impact on the average weighted impact score. Nevertheless, physical activity was positively related to this measure of QoL ($p = 0.010$). Furthermore, using the LADA patients undergoing insulin treatment with or without DR as reference groups, we could estimate the combined coefficients using the same model (Table S1). These analyses showed that LADA subjects with DR who were treated with insulin showed a lower QoL than any other combination of diabetes type, insulin treatment and DR. Furthermore, the LADA patients undergoing insulin treatment without DR had a lower QoL than the T2DM patients without insulin treatment either with ($p = 0.006$) or without DR ($p < 0.001$).

**Table 4  Multivariate linear regression for the Audit of Diabetes Dependent Quality of Life (ADDQoL) average weighted impact score.**

| Coefficients | Estimate | Standard error | *p* value |
|---|---|---|---|
| Intercept | −1.29375 | 0.21782 | <0.001 |
| T2DM* without insulin | 0.95200 | 0.20699 | <0.001 |
| LADA* without insulin | 1.06176 | 0.64037 | 0.100 |
| T2DM* insulin | 0.10006 | 0.31884 | 0.750 |
| T1DM | −0.04103 | 0.22328 | 0.850 |
| DR | −1.25952 | 0.40942 | 0.002 |
| Disease duration | −0.01674 | 0.00663 | 0.010 |
| No education** | −0.45008 | 0.15383 | 0.004 |
| Physical activity | 0.27361 | 0.10539 | 0.010 |
| Smoker, current | −0.21031 | 0.12937 | 0.100 |
| Smoker, former | −0.30882 | 0.11674 | 0.008 |
| T2DM* without insulin* DR | 0.92912 | 0.43927 | 0.030 |
| LADA* without insulin* DR | 1.10620 | 1.05301 | 0.290 |
| T2DM* insulin* DR | 1.23910 | 0.49206 | 0.010 |
| T1DM* DR | 0.98320 | 0.44443 | 0.030 |

**Notes.**
Multiple R-squared: 0.2895 (six cases with missing information for any variable in the model).
Reference group: LADA patients receiving insulin treatment.
*Indicates the existence of interactions between variables.
**"No education" identifies patients who did not complete compulsory education.
LADA, latent autoimmune diabetes of adult; T2DM, type 2 diabetes mellitus; T1DM, type 1 diabetes mellitus; DR, diabetic retinopathy.

## Treatment satisfaction

The proportion of subjects with a lower DTSQ final score differed among the study groups: LADA, 60.4%; T2DM, 52.5%; and T1DM, 41.9% ($p = 0.049$). However, individual paired comparisons between the groups did not yield statistically significant differences (Table 5). The multivariate linear regression analysis of the DTSQ final score revealed no differences between the different combinations of groups according to diabetes type and insulin-treatment (Table S2). Physical activity had a positive impact ($p = 0.001$) and former smoking had a negative impact on the DTSQ final score ($p = 0.01$).

Concerning another measure of TS, the proportion of patients with a perception of increased hyperglycemia frequency was higher in the LADA group (87.5%) than in the T2DM (53.9%) and T1DM groups (71%; $p < 0.001$ and $p = 0.039$, respectively). The multivariate linear regression analysis found an interaction between the type of diabetes and insulin treatment (Table S3). The insulin-treated LADA patients had a higher hyperglycemia frequency perception than the T1DM ($p = 0.04$) and insulin-treated T2DM subjects ($p = 0.05$). Physical activity was associated with a lower hyperglycemia perception frequency ($p = 0.002$). Additionally, DR and higher HbA1c were associated with higher hyperglycemia frequency perception ($p = 0.05$ and $p < 0.001$, respectively).

Finally, the proportion of patients with higher hypoglycemia frequency perception differed among the groups ($p < 0.001$; Table 5). A higher proportion of patients with T1DM (55.6%) than patients with LADA (33.3%) reported a high frequency of perceived

**Table 5  Results of the variables corresponding to the Diabetes Treatment Satisfaction Questionnaire (DTSQ) in the different study groups.**

| Items | LADA (N = 48) n (%) | T2DM (N = 297) n (%) | T1DM (N = 124) n (%) | p-value* | p-LADA vs. T2DM | p-LADA vs. T1DM |
|---|---|---|---|---|---|---|
| Hyperglycemia frequency perception | | | | <0.001 | <0.001 | 0.039 |
| 0–2 | 6 (12.5) | 137 (46.1) | 36 (29) | | | |
| 3–6 | 42 (87.5) | 160 (53.9) | 88 (71) | | | |
| Hypoglycemia frequency perception | | | | <0.001 | 0.095 | 0.021 |
| 0–2 | 32 (66.7) | 234 (78.8) | 55 (44.4) | | | |
| 3–6 | 16 (33.3) | 63 (21.2) | 69 (55.6) | | | |
| Final score | | | | 0.049 | 0.389 | 0.091 |
| [4,28] | 29 (60.4) | 156 (52.5) | 52 (41.9) | | | |
| [28,36] | 19 (39.6) | 141 (47.5) | 72 (58.1) | | | |

**Notes.**
The groups are stratified by medians.
*p-value for comparisons between groups.
LADA, latent autoimmune diabetes of adult; T2DM, type 2 diabetes mellitus; T1DM, type 1 diabetes mellitus.

hypoglycemia ($p = 0.021$). In the multivariate linear regression analysis (Table S4), T2DM patients without insulin treatment showed a significantly lower hypoglycemia frequency perception compared with insulin-treated LADA subjects ($p = 0.006$). Additionally, the following conditions increased this measure: T1DM ($p = 0.02$), female sex ($p = 0.001$) and disease duration ($p = 0.006$).

## DISCUSSION

In the current study, we demonstrated that LADA patients presented with lower diabetes-associated specific and average weighted impact QoL scores than patients with T2DM; however, we found no differences in terms of present QoL. Insulin treatment had a negative impact on diabetes-specific QoL, and the subgroup of insulin-treated patients with LADA did not differ from other insulin-treated groups (T2DM and T1DM) in this regard; however, the average weighted impact score was poorer in insulin-treated LADA subjects than in their corresponding T2DM counterparts. The group with the worst values for this impact score was the subgroup of LADA patients with DR and insulin treatment.

Although there was a significant difference in the DTSQ final score among the groups, paired comparisons between groups did not reach significant differences; however, there were differences concerning the hyperglycemia and hypoglycemia frequency perception among the groups. The LADA patients showed an increased hyperglycemia frequency perception compared with the T2DM and T1DM groups, which was mainly at the expense of the insulin-treated LADA group. In contrast, the LADA patients had an improved hypoglycemia perception frequency compared with the T1DM patients.

To our knowledge, there are no previous studies on QoL and TS in subjects with LADA. Concerning quality of life, previous studies revealed a lower QoL in patients with type 1 diabetes or insulin-treated type 2 diabetes and in subjects with one or more late diabetic

complications (*Collins et al., 2009*). In a previous study by our group, we also found that patients with T2DM and DR had a lower QoL than those without this complication (*Alcubierre et al., 2014*), which is in line with the current findings in patients with LADA. Our results are also in line with other previous studies that reported that insulin therapy and diabetic complications were associated with a poorer QoL in patients with T2DM (*Bradley & Speight, 2002*; *Collins et al., 2009*; *Depablos-Velasco et al., 2014*; *Shim et al., 2012*; *Speight & Bradley, 2000*; *Sundaram et al., 2007*); however, in a longitudinal study of patients with T2DM starting insulin therapy, QoL improved six months after the commencement of insulin therapy (*Wilson, Moore & Lunt, 2004*). The negative impact of insulin treatment on LADA patients is associated with a poorer QoL perception, which may be linked to impaired metabolic control and the delayed initiation of insulin treatment. Furthermore, *Depablos-Velasco et al. (2014)* observed that diabetic patients with poor metabolic control had a low QoL; however, we found no significant differences in terms of glycemic control.

*Shim et al. (2012)* described that the ADDQoL average weighted impact score was lower in association with male gender, higher education level and longer disease duration. Other studies have related lower QoL to advanced age, female sex, lower educational level, obesity, the presence of diabetic comorbidities, poorer glycemic control and lower socioeconomic status in the type 1 and type 2 diabetic populations (*Ahola et al., 2010*; *Bradley & Speight, 2002*; *Collins et al., 2009*; *Imayama et al., 2011*; *Nicolucci et al., 2009*; *Oliva, Fernandez-Bolanos & Hidalgo, 2012*; *Shim et al., 2012*; *Speight & Bradley, 2000*; *Sundaram et al., 2007*). These findings are similar to our results concerning the presence of complications (i.e., DR), longer disease duration and waist circumference, all of which were related to a lower QoL.

However, we found a positive relationship between QoL and physical activity in patients with diabetes. Physical activity is an important component of the lifestyle measures used to treat patients living with diabetes. *Imayama et al. (2011)* performed a longitudinal study of 490 T1DM and 1,147 T2DM patients to investigate the determinants of health-related QoL (HRQoL). The authors also found a higher HRQoL in patients with a high physical activity level.

Concerning TS, the results showed a relatively high score despite the negative impact of diabetes on QoL, which has been reported in previous studies (*Bradley & Speight, 2002*; *Speight & Bradley, 2000*; *Speight, Reaney & Barnard, 2009*). We could not detect any difference between the LADA patients and the other groups in the DTSQ final score. Nevertheless, the hyperglycemia frequency perception was worse in the insulin-treated LADA patients, although glycemic control was not different between the groups. The increased hyperglycemia frequency perception in the subjects with LADA may be attributable to previous poorer glycemic control compared with type 1 diabetic patients under stable control. It is worth noting that the LADA subjects were recruited from a specialized hospital clinic, where patients are usually referred from primary care because of poor glycemic control. In contrast, the frequency of hypoglycemia perception was similar between the LADA and T2DM patients and was clearly increased in T1DM patients. This finding could be attributed to a higher intensity of insulin treatment in T1DM, which may be associated with a higher frequency of hypoglycemia in these patients. Unfortunately, the frequency of previous mild and severe hypoglycemia episodes was not

assessed. Additionally, as expected from the exposure to insulin treatment, the insulin-treated LADA patients had an increased frequency of hypoglycemia compared with the non-insulin treated T2DM patients.

Improved TS has been associated with optimal glycemic control (HbA1c $\leq$ 7%) in T2DM without concomitant complications or insulin treatment (*Biderman et al., 2009*; *Mancera-Romero et al., 2016*). These findings are in line with those reported here. Furthermore, we found that physical activity showed a positive association with TS, as in our previous study that involved the subgroup of T2DM patients included in our study (*Alcubierre et al., 2014*); however, we could not identify other studies addressing this specific issue in the diabetic population.

The current study has several limitations. A causal relationship between QoL, TS and related factors could not be established because of the cross-sectional study design; however, the variation in the QoL among patients was shown to be strongly influenced by the characteristics that do not vary over time (*Alva et al., 2014*). The low number of patients with LADA that were included is an important limitation. Therefore, the low number of subjects in the LADA group raises a point of caution regarding the external validity of the current results. Additionally, a significant number of potential subjects from the local LADA cohort refused to participate. However, there were no differences between the participating LADA patients and the non-participating LADA patients in the proportion of comorbidities that could affect the main study outcomes. Thus, the current study should be considered an exploratory investigation that raises awareness of the need for further studies of patient-oriented outcomes in subjects with LADA. Additionally, the current study compared the QoL and TS of LADA patients with that of patients with the two main classical types of diabetes. Another limitation arises from the potential selection bias of the current study hospital setting. Subjects with worse glycemic control are referred to specialized care for further diagnostic work-up that leads to the final diagnosis of LADA, while subjects with LADA who have better glycemic control are likely to remain unidentified at the primary care level. Therefore, the current results may not be extrapolated to the whole population of LADA subjects. The issue of poor QoL is very relevant to patients with LADA who are ultimately referred to specialized care because of unstable glycemic control and the need for insulin treatment. Finally, mental well-being may have an impact on the main outcomes evaluated in the current study. Although mood items are included in the ADDQoL-19 questionnaire, a proper evaluation of emotional or mental well-being was not performed in this study. This latter issue should be taken into consideration in future studies of patients with LADA.

## CONCLUSIONS

In conclusion, in the current study we found that that LADA patients with DR who were undergoing insulin treatment had a negative QoL compared with T2DM and T1DM patients. Furthermore, the LADA patients undergoing insulin treatment perceived a greater frequency of hyperglycemia than the other diabetic groups. Further research is warranted to study the status and changes over time in the QoL and TS of patients with LADA in other settings and with a larger number of patients.

### Funding

This study was supported by grant PS09/01035 and PI12/00183 from the Instituto de Salud Carlos III, Ministry of Economy and Competitiveness, Spain. CIBERDEM is an initiative from Instituto de Salud Carlos III (Plan Nacional de I+D+I and Fondo Europeo de Desarrollo Regional). MGC holds a predoctoral fellowship from the Ministerio de Educación, Cultura y Deporte (FPU15/03005). The funders had no role in study design, data collection and analysis, decision to publish, or preparation of the manuscript.

### Grant Disclosures

The following grant information was disclosed by the authors:
Instituto de Salud Carlos III, Ministry of Economy and Competitiveness: PS09/01035, PI12/00183.
Ministerio de Educación, Cultura y Deporte: FPU15/03005.

### Competing Interests

The authors declare there are no competing interests.

### Author Contributions

- Minerva Granado-Casas conceived and designed the experiments, performed the experiments, analyzed the data, contributed reagents/materials/analysis tools, wrote the paper, prepared figures and/or tables.
- Montserrat Martínez-Alonso analyzed the data, contributed reagents/materials/analysis tools, prepared figures and/or tables.
- Nuria Alcubierre, Anna Ramírez-Morros, Marta Hernández and Esmeralda Castelblanco performed the experiments, contributed reagents/materials/analysis tools.
- Joan Torres-Puiggros contributed reagents/materials/analysis tools, prepared figures and/or tables, reviewed drafts of the paper.
- Didac Mauricio conceived and designed the experiments, analyzed the data, contributed reagents/materials/analysis tools, wrote the paper, prepared figures and/or tables, reviewed drafts of the paper.

### Human Ethics

The following information was supplied relating to ethical approvals (i.e., approving body and any reference numbers):

The study was approved by the Ethics Committee of University Hospital Arnau of Vilanova from Lleida.

### Data Availability

The raw data and code have been provided as a Supplemental File.

## Supplemental Information

Supplemental information for this article can be found online at http://dx.doi.org/10.7717/peerj.3928#supplemental-information.

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
