# Peer review of "Decreased quality of life and treatment satisfaction in patients with latent autoimmune diabetes of the adult"

_PeerJ, doi:10.7717/peerj.3928_

## Round 0.1 · original submission · Major Revisions

Dear Authors,

The paper seems of interest, but it needs some major revisions. Please answer to the reviewers points.

Regards

·

Basic reporting

The article is written in a clear English.
The introduction and the background are complete and appropriately referenced
to relevant prior literature.
All results relevant to the hypothesis are showed in the text and in the tables.

Experimental design

The research question is clear, relevant and meaningful and the study is rigorous, well detailed and conducted according to technical standard.
Validated tools were used in the language in which the study was conducted.
However, it would be appropriate to enclose questionnaires as supplementary materials, in order to facilitate a better understanding of the items.

Validity of the findings

The study, showing that LADA patients have a poorer QoL profile than patients with T1DM and T2DM, especially at the expense of DR and insulin treatment, provides new results on a topic that is currently little explored.

However, the study included patients with chronic pathology with a view to assessing the impact on quality of life. Notably, quality of life is dependent on mental well-being. Exclusion criteria include mental illness. Therefore, as the mood of these patients was evaluated?
Would it not be more appropriate to include this variable in statistical analyses to assess its impact on quality of life and Treatment Satisfaction?
Another question.
Another exclusion criterion is the presence of dementia. Who and how did patients evaluate to exclude cognitive impairment?

Reviewer 2 ·

Basic reporting

No comment

Experimental design

No comment

Validity of the findings

1) The most important issue is the limited number of LADA patients who agreed to participate in the study: 86 contacted patients and only 48 participants, with a refusal frequency equal to 44% . This high proportion of refusal is an important selection bias that can impact on results validity.
Which factors influenced participation in the study? Can authors exclude that participation was influenced by disease status, socio-demographic factors, quality of life or treatment satisfaction? Which are the differences beetween patient who partecipated and those who refused?
This point is crucial for results validity!

2) Additionally, given the high number of statistical tests, a Bonferroni correction should be applied.

Reviewer 3 ·

Basic reporting

1. Your MATERIALS AND METHODS needs more details about the inclusion criteria for LADA,T1DM,and T2DM, respectively,especially for the inclusion criteria for the LADA.
2. I suggest that the results you describing in Lines 165-177 shoud be moved forward to the part of MATERIALS AND METHODS.
3.It is noted that your manuscript needs careful editing by someone with expertise in technical English editing paying particular attention to English grammar, spelling, and sentence structure.

Experimental design

no comment

Validity of the findings

Lines 237-239:T2DM patients without insulin treatment showed a significantly lower hypoglycemia frequency perception in comparison with insulin-treated LADA subjects (p=0.006).Please explain why ?
Lines 228-229:Insulin-treated LADA patients had a higher hyperglycemia frequency perception than T1DM (p=0.04)...... Please explain why ?

Additional comments

The number of subjects with LADA is too small so that the conclusion that "LADA patients showed a poorer QoL profile than patients with T1DM and T2DM" in the current study was not convincing.

---

## Round 0.2 · accepted · Accept

I'm glad to inform you that the paper is now suitable for PeerJ.

·

Basic reporting

The manuscript is well structured and adequate regarding literature references.
It contains relevant results to support the hypothesis.

Experimental design

Research question is well defined, relevant and meaningful.
Methods are well described.
Limitations are adequately described in discussion section.

Validity of the findings

Results are significant, but partially inconclusive, suggesting the need for further studies on this issue.

Additional comments

It is worth appreciating the attempt to explore what influences the quality of life of patients with LADA, even if the impact that mood alterations may have on the management and perception of the disease is only partially reported. This is a study limitation. However, this point only partially limits the success of the study which is adequate and complete.